# Sensors on Internet of Things Systems for the Sustainable Development of Smart Cities: A Systematic Literature Review

**DOI:** 10.3390/s24072074

**Published:** 2024-03-24

**Authors:** Fan Zeng, Chuan Pang, Huajun Tang

**Affiliations:** School of Business, Macau University of Science and Technology, Taipa, Macao 999078, China; fzeng@must.edu.mo (F.Z.); cpang@must.edu.mo (C.P.)

**Keywords:** sensors, internet of things, sustainable development, smart city

## Abstract

The Internet of Things (IoT) is a critical component of smart cities and a key contributor to the achievement of the United Nations Sustainable Development Goal (UNSDG) 11: Sustainable Cities and Communities. The IoT is an infrastructure that enables devices to communicate with each other over the Internet, providing critical components for smart cities, such as data collection, generation, processing, analysis, and application handling. IoT-based applications can promote sustainable urban development. Many studies demonstrate how the IoT can improve smart cities’ sustainable development. This systematic literature review provides valuable insights into the utilization of the IoT in the context of smart cities, with a particular focus on its implications for sustainable urban development. Based on an analysis of 73 publications, we discuss the role of IoT in the sustainable development of smart cities, focusing on smart communities, smart transportation, disaster management, privacy and security, and emerging applications. In each domain, we have detailed the attributes of IoT sensors. In addition, we have examined various communication technologies and protocols suitable for transmitting sensor-generated data. We have also presented the methods for analyzing and integrating these data within the IoT application layer. Finally, we identify research gaps in the literature, highlighting areas that require further investigation.

## 1. Introduction

Cities are the hubs of economic, social, cultural, and everyday life [1]. Currently, the urban population is 4.27 billion, making up approximately 55% of the global population [1,2]. Projections indicate that by 2050, around 70% of the world’s population will reside in cities, significantly expanding urban areas by 1.2 million km^2^ [2].

Despite being centers of activity, cities often grapple with higher population densities, posing challenges in providing adequate services and infrastructure for their residents [3]. Rapid urbanization and industrialization have hindered sustainable development, leading to issues such as pandemics, severe environmental pollution, inefficient waste management, strained power grids, traffic congestion, and the uneven distribution of education and medical resources [4]. For instance, cities currently account for 60% of energy consumption and over 75% of greenhouse gas emissions globally [5]. Smart cities present a promising solution to tackle these challenges, including epidemic control, population growth, environmental pollution, and energy shortages [6]. For example, Yang and Chong [7] found that COVID-19 would be more controllable in smart cities. Generally speaking, a smart city is defined as “an urban environment that utilizes information and communication technology (ICT) and other related technologies to enhance performance efficiency of regular city operations and quality of services provided to urban citizens” ([6], p. 697).

A crucial component of smart cities is the Internet of Things (IoT), an infrastructure enabling devices to communicate over the Internet [3]. The IoT provides critical components for smart cities, such as data collection, data analysis, and application handling [6]. By harnessing IoT capabilities, smart cities can gather and analyze data on climate, emissions, traffic, waste management, and disasters [8]. Utilizing these data, smart cities can develop various applications to improve transportation, urban infrastructure, resource utilization, urban life, governance, and the overall environment [9]. Consequently, the IoT contributes to the sustainable development of smart cities and enhances the quality of life for their citizens [10]. Effectively integrating IoT technology into the construction of smart cities is a critical aspect of building a developed and sustainable society [10]. Thus, IoT plays a pivotal role in helping smart cities achieve the United Nations’ Sustainable Development Goal (UNSDG) 11, “Sustainable Cities and Communities” [9].

Lots of practical examples show that the IoT is able to improve the sustainable development of smart cities. With IoT technology, Santander City and Nice City can use sensors to collect data on available parking lots to implement smart parking management systems [11,12]. The IoT-based smart lighting system in Padova city can monitor lighting periodically or upon request based on the light intensity data collected from sensors, which saves energy, reduces maintenance costs, and improves service quality [6].

The literature has shown a substantial research interest in the application of IoT in smart cities, leading to several existing reviews, such as those by Bellini et al. [13], Blasi et al. [1], Ramírez-Moreno et al. [14], and Silva et al. [6]. However, there remains a scarcity of comprehensive understanding regarding the role of IoT in fostering sustainable development within smart cities. Therefore, this study aims to contribute valuable insights into the utilization of IoT in the context of smart cities, with a particular focus on its implications for sustainable urban development and bridging the knowledge gap. Therefore, we develop the following research questions:

What types of sensors are used to collect data? What are the different aspects of smart city sustainability?

We aim to discuss current global best practices on how IoT systems can be used for smart city sustainability and provide a comprehensive analysis of the sensors used in these IoT systems in different aspects of smart cities.

2.What communication technologies and protocols are used to transmit data from sensors?3.What methods are used to analyze sensor data?

Additionally, this paper will conclude by highlighting the issues that have been neglected in the application of IoT by smart cities to their sustainability. Possible opportunities for future research in this area will also be explored.

## 2. Literature Review

### 2.1. Internet of Things

The IoT can be defined as “a worldwide network of interconnected objects uniquely addressable, based on standard communication protocols” ([15], p. 685). A more comprehensive description characterizes IoT as “an ecosystem that contains smart objects equipped with sensors, networking and processing technologies integrating and working together to provide an environment in which smart services are taken to the end users” ([16], p. 241). The IoT finds widespread application across various domains, with predictions indicating that by 2030, there will be 30 billion IoT devices deployed worldwide [17]. These ubiquitous IoT devices form an interconnected structure resembling the future Internet, influencing not only cities but also the world at large [18]. Smart cities, energy conservation, pollution control, smart homes, emission reduction, intelligent transportation, and the smart industry have all experienced transformations through the integration of IoT [19].

The architecture of the IoT mainly consists of three layers: the perception layer, the network layer, and the application layer [20]. The perception layer, as the foundational level of IoT, comprises physical devices such as sensors that collect data and transmit it to the network layer [19]. The network layer manages the reception and transportation of substantial volumes of sensor-produced data to the application layer through various communication technologies and protocols, facilitated by gateways [21]. At the application layer, data received from the network layer is stored in databases, and various tools are employed for data analysis, leading to the generation of diverse applications related to smart cities, smart health, and other fields [13].

### 2.2. Internet of Things and the Sustainable Development of the Smart City

A smart city is defined as “an urban environment that utilizes information and communication technology and other related technologies to enhance performance efficiency of regular city operations and quality of services provided to urban citizens” ([6], p. 697). Smart cities can promote sustainable development through various smart solutions, such as smart grids, smart factories, smart warehouses, smart communities, smart healthcare, smart transportation, and smart hospitals [6]. For instance, the implementation of a smart grid in Helsinki resulted in a 15% reduction in energy consumption [22]. Smart healthcare systems can record cases and monitor epidemic trends, as demonstrated during the COVID-19 pandemic [13]. Additionally, smart mobility services like Jelbi in Berlin, which integrate various transportation options, aim to foster a transition towards more sustainable mobility [13]. The sustainable development of smart cities encompasses most of the UNSDGs and ultimately contributes to achieving SDG 11, “Sustainable Cities and Communities” [9,13].

The IoT plays a vital role in the realization of these smart solutions. The IoT is the main component in many smart cities, such as Singapore, London, and Helsinki [10]. By monitoring parameters like energy consumption, water quality, and traffic flow through IoT, smart cities can proactively anticipate risks and implement preventive measures, thereby enhancing sustainability [21]. The IoT connects billions of sensor devices to the Internet, offering advantages such as cost-effectiveness, ease of use, and real-time information retrieval [5]. Consequently, the IoT is widely used in environmental management, allowing the detection and evaluation of changes in pressure, gas levels, and temperature, among other factors [5]. For example, London installed 100 IoT sensors in 2019 to monitor air quality [10]. In 2017, the University of Helsinki monitored air quality with an IoT system [10]. Furthermore, the data collected through IoT facilitates the efficient distribution of energy and public resources. For instance, Singapore utilizes IoT technology to manage solid waste through an integrated waste management facility, improving treatment efficiency and reducing greenhouse gas emissions [23]. Furthermore, IoT greatly increases the potential for disaster detection and disaster risk analysis, enabling rapid response to disasters [24]. Many practical examples show that the IoT is effective in monitoring natural disasters, sending warnings, and alerting disaster management authorities [24]. In 2020, 23 UN countries had efficient disaster early warning systems, which protected 93.63% of the population at risk of natural disasters in these countries successfully [25]. Therefore, the IoT can help smart cities improve environmental sustainability.

In addition, IoT can also help smart cities improve social sustainability. The IoT helps create a society that citizens trust [10]. Centrist citizen orientation is a common feature of the IoT systems in Singapore, London, and Helsinki [10]. If citizens can access information transparently from the city’s website and information panel, they will develop trust and engagement in the city [10]. Furthermore, the IoT also can improve the security of smart cities. For example, Singapore, which installed sensor cameras on its streets to identify facial features, reported no thefts or robberies in 322 days in 2018 [10]. What is more, IoT-supported smart healthcare can help hospitals monitor patients’ physiological parameters for early detection of clinical deterioration [26]. Additionally, IoT technology has the potential to communicate instant information updates and can become a key player in disaster relief activities [27]. The dynamic nature of the requirements and environment in rescue operations emphasizes the ability to make effective and precise decisions in the shortest possible time [27]. The first 72 h after a disaster (the golden hour of rescue) are critical for search and rescue, as the probability of finding survivors drops dramatically after this time [28]. The IoT can improve search and rescue efficiency within 72 h by detecting vital signs and locating trapped personnel [29].

## 3. Methodology

The methodology employed in this paper primarily relies on a systematic literature review (SLR). SLR plays a pivotal role in defining the scope of the review by synthesizing data and mapping the existing literature in the research area, thereby aiding in the development of frameworks that integrate current knowledge [30,31].

The SLR process typically involves four main steps, including determining keywords and the inclusion and exclusion criteria, conducting a comprehensive search of relevant publications, filtering these publications based on the established criteria, and finally, discussing and analyzing the finalized set of publications [32].

We collected publications from the Web of Science (WoS) database only since lots of high-quality review papers also collected publications from a single database [33,34,35]. We conducted a keyword search on WoS to find the target publications on 22 December 2023. We grouped the keywords into three parts: (1) IoT keywords [25,36]; (2) the keywords related to “sustainable”; and (3) the names of top 100 smartest cities according to the IMD Smart City Index 2023 (https://www.imd.org/wp-content/uploads/2023/04/smartcityindex-2023-v7.pdf?trk=public_post_comment-text accessed on 1 December 2023). Appendix A presents the top 100 smart cities in the IMD Smart City Index 2023 list. Therefore, our final search string is “TS=(“internet of thing*” OR “IoT” OR “IoTs” OR “wireless sensor network*” OR “WSN” OR “WSNs” OR “Internet of Everything” OR “IoE”) AND TS=(sustainab* OR green* OR Resilien* OR environment* OR social* OR responsib* OR eco-friendly) AND TS=(“Zurich” OR “Oslo” OR “Canberra” OR “Copenhagen” OR “Lausanne” OR “London” OR “Singapore” OR “Helsinki” OR “Geneva” OR “Stockholm” OR “Hamburg” OR “Beijing” OR “Abu Dhabi” OR “Prague” OR “Amsterdam” OR “Seoul” OR “Dubai” OR “Sydney” OR “Hong Kong” OR “Munich” OR “New York” OR “Auckland” OR “Wellington” OR “Brisbane” OR “Shanghai” OR “Reykjavik” OR “Bilbao” OR “Vienna” OR “Taipei” OR “Riyadh” OR “Melbourne” OR “Tallinn” OR “Berlin” OR “Boston” OR “Brussels” OR “Gothenburg” OR “Madrid” OR “Dusseldorf” OR “Washington” OR “Ottawa” OR “Rotterdam” OR “Vancouver” OR “Hague” OR “Warsaw” OR “Luxembourg” OR “Paris” OR “Ljubljana” OR “Toronto” OR “Busan” OR “Los Angeles” OR “Bologna” OR “Mecca” OR “Denver” OR “Zaragoza” OR “Seattle” OR “Jeddah” OR “Hanover” OR “Nanjing” OR “Doha” OR “Zhuhai” OR “Chicago” OR “Bratislava” OR “Dublin” OR “Lyon” OR “Vilnius” OR “Shenzhen” OR “Tianjin” OR “San Francisco” OR “Montreal” OR “Hangzhou” OR “Guangzhou” OR “Tokyo” OR “Manchester” OR “Birmingham” OR “Barcelona” OR “Leeds” OR “Newcastle” OR “Bordeaux” OR “Krakow” OR “Glasgow” OR “Kiel” OR “Milan” OR “Riga” OR “Lille” OR “Medina” OR “Chongqing” OR “Budapest” OR “Bangkok” OR “Kuala Lumpur” OR “Ankara” OR “Tel Aviv” OR “Philadelphia” OR “Phoenix” OR “Cardiff” OR “Belfast” OR “Muscat” OR “Chengdu” OR “Osaka” OR “Lisbon” OR “Hanoi”)”. In addition, we only consider English peer-reviewed studies. Finally, we used the filters in WoS to identify the studies that meet UNSDG 11, “Sustainable Cities and Communities”, resulting in 255 publications.

Then, we screened them and removed the irrelevant articles. The selection process consists of two stages. During the first stage, publications related to the application of IoT in the sustainable development of smart cities are selected through screening of their titles, abstracts, and keywords. We retained 164 publications after the selection in the first stage. In the second stage, the full text of each selected publication was read to ensure it fell within the scope of our focus. Furthermore, we only considered studies connected to the real world which could accelerate the application of the research findings. In other words, scholars who applied their research results to the real-world environment of a smart city or utilized the data generated by sensors in a smart city for data analysis. Finally, we obtained 73 publications.

Firstly, we conducted a descriptive analysis of 73 publications to present the trend of “the sustainable development of smart cities”. The descriptive analysis included the publication number distribution by year and by source.

Secondly, to figure out the major topics related to the sustainable development of smart cities, we conducted a thematic analysis through reading the abstracts of the publications. The categorization was also revised while reading the full text of the publication. For thematic analysis, researchers usually categorize the data into themes based on recurring ideas.

Thirdly, we conducted a content analysis for each included publication. We systematically analyzed the full text of each publication to identify the meanings within the content. We summarized key findings and recorded the basic information on Appendix B. Appendix B presents each included publication’s information, such as sensor type, application, sustainability measure, challenge, and outcome. However, we may only record sustainability measures, challenges, and outcomes of the publications if they solely focus on the analysis and application of sensor-generated data. We synthesize the findings to identify the pattern and trend. For the sample publications on the same topic, we discuss them in chronological order so as to facilitate understanding of the technology’s development. Then, we discuss discrepancies and inconsistencies and write a critical appraisal of the findings. We also explain the heterogeneity among included publications, such as differences in methodologies, environment, and the construction of IoT systems.

Finally, we conducted a framework analysis to integrate the IoT systems for different aspects of sustainable development of smart cities.

## 4. Descriptive Analysis

Table 1 displays the top six publication sources. Among the 73 publications, most were published in the journal *Sensors* (7 publications), followed by *IEEE Internet of Things Journal* (5 publications).

Figure 1 shows the distribution of publications by year. Scholars started to discuss the role of IoT in the sustainable development of smart cities in 2008. The number of publications fluctuated upward over the next 12 years, peaking in 2020 with 14 publications. The dataset of articles for 2023 is still incomplete, as it only includes items downloaded until 22 December 2023.

Among the 73 publications, 35 publications discussed smart communities. Twenty-three publications focused on smart transportation. Seven publications focused on disaster management. Two publications highlighted the privacy and security of the IoT system applied in the sustainable development of smart cities. The remaining six publications are related to some topics, such as smart agriculture (two publications), smart healthcare (two publications), smart education (one publication), and smart tourism (one publication). Figure 2 presents the distribution of publications by topics.

## 5. Smart Community

Smart communities converge smart environment management and smart waste management [6], which meets SDG 11.6: “By 2030, reduce the adverse per capita environmental impact of cities, including by paying special attention to air quality and municipal and other waste management” (https://sdgs.un.org/goals/goal11#targets_and_indicators, accessed on 10 February 2024).

### 5.1. Smart Environment Management

Smart environment management includes air quality monitoring, noise monitoring, and soil monitoring. The IoT system can monitor the environment in real time around the clock. It can also collect long-term environmental data, which can then be analyzed to help citizens understand environmental trends and identify possible sources of pollution.

#### 5.1.1. Air Quality Monitoring

Air quality monitoring involves managing both the interior and exterior air quality monitoring.

Internal air quality monitoring focuses on monitoring humidity and temperature to provide a suitable environment for human beings and special objects, such as books in a library. To improve the performance of the built environment in order to provide comfortable living environments and improve energy efficiency, Hossain et al. [37] introduced a low-cost, high-resolution IoT system at the University of Westminster’s Marylebone Campus in central London. This system can monitor environmental parameters such as carbon dioxide, dry-bulb temperature, illuminance level, relative humidity and sound levels. By detecting these environmental factors, the IoT system can intelligently adjust the operation of indoor air conditioning equipment to ensure environmental comfort and save energy consumption. To protect cultural collections and prevent the risk of manual data collection, Grossmann et al. [38] developed MonTreAL, a library-wide IoT system for monitoring humidity and temperature. Further, Grossmann et al. [39] expanded MonTreAL and introduced SensIoT, a general sensor monitoring framework for the IoT. Its most prominent feature is the use of Docker and Docker Swarm to solve common software problems and enable flexible data collection. The SensIoT system is an open-source sensor monitoring system that enables easy deployment and maintenance while remaining flexible and scalable. Martínez et al. [40] developed an IoT system to monitor energy consumption through the measurement of CO_2_ levels, temperature, and humidity. The measurement of CO_2_ levels, temperature, and humidity helps to establish air conditioning schedules and improve a building’s heating, ventilating, and air conditioning. They use sensors to measure temperature [38,39] and humidity [38,39]. The sensor model DHT-22 can both record temperature and humidity [38]. Other humidity sensors are ASH2200 [39], DHT11 [39], AM2302 [39]. The IoT system continuously monitors changes in humidity and temperature to keep cultural collections in a suitable environment throughout the day for long-term preservation.

External air quality monitoring is mainly focused on urban transportation. Liu et al. [41] deployed an IoT air quality monitoring system on the main roads in Taipei to monitor carbon monoxide concentrations caused by vehicle emissions. Similarly, Wen et al. [42] deployed IoT systems at crossroads and main roads in Taipei to monitor carbon monoxide levels in urban areas. Hoang et al. [43] used an IoT system to monitor air pollution levels caused by transportation in Hanoi. They visualized the data using calibration and data clustering techniques, mathematical interpolation methods, and computer graphics. In recent years, low-cost IoT systems for air quality monitoring have become one of the new trends. Wilhelm et al. [44] developed a low-cost and wearable IoT system to collect environmental data robustly. Daepp et al. [45] also developed a low-cost IoT system, Eclipse, to monitor Chicago’s outdoor air quality and found that the system reliably collected data for more than 90% of the expected hours. Last but not least, Garau et al. [46] reported on IoT Environmental Monitoring systems in Florence, Helsinki, and Cagliari.

For external air quality monitoring, people usually use sensors to collect air quality data and weather data to build up a more comprehensive and accurate IoT system for air quality monitoring. Table 2 presents the parameters collected for air quality data. Table 3 presents the parameters collected for weather data. PM concentration seems to be the main monitoring indicator. Recently, focus has been more on the cost of the sensor. Scholars have begun to utilize low-cost sensors to collect air data. Cowell et al. [47] proved that Plantower, Nanchang, China, PMS5003 sensor, a kind of low-cost optical particle counter, is effective to collect PM concentration, which responds to questions about the reliability of low-cost PM sensor data. The Plantower, Nanchang, China, PMS5003 sensor is an optical particle counter that uses light scattering techniques to determine the number and size of particles [47]. Dust build-up would be the primary cause of failure for particle counters [48]. To solve this issue, Van Kessel et al. [48] introduced an improved PM sensor, Minimum Airflow Particle Counter. In addition, the Beta ray analyzer is another type of PM sensor that measures PM concentration by the differences in radiation strength on the filter paper [49]. As for CO sensors, they typically monitor CO concentration through fluctuations in conductivity, such as the MiCS-5525 model [42]. In addition, they may also be chemical sensors (e.g., City Technology Ltd., Portsmouth, UK, A3CO sensor) [50]. For the calibration of chemical sensors, Arfire, Marjovi, andMartinoli [50] proposed a model-based sliding window single-hop rendezvous calibration algorithm to estimate the baseline and gain characteristics of chemical sensors, taking into account temperature dependencies and temporal drift of the sensors.

Collecting several air and/or weather parameters with one sensor seems to be a trend nowadays in order to save costs and materials. For example, Aranet, Riga, Latvia, Aranet 4 Pro, a model of CO_2_ sensor, can measure temperature, humidity, and barometric pressure [40]. Weather stations (e.g., La Crosse Technology, Wisconsin, USA, WS-2316 [41]) can collect several types of weather data, including barometric pressure, relative humidity, temperature, rainfall, wind direction, and wind speed. The BME280 sensor also can measure temperature, humidity, and barometric pressure together [51]. Common temperature and humidity sensor models include Sensirion, Stäfa, Switzerland, SHT11, SHTC3, and Si7021-A20-GM1 [41,44,45].

IoT platforms used to install sensors should be able to process the data from the sensors with low power consumption [59]. They include Texas Instruments MSP430F1611 [41], Triton XXS [60], Arduino Nano [52], and Arduino [44]. The processors include Arduino MKRFox1200 [47], NRF9160 [45], Atmel ATmega8 [44], NodeMCU-32S [44], and STM32 F103 [61]. The single computer boards include the Texas Instruments Waswpmote platform [43], Libelium [57], Raspberry PI [37,52,62], and ARK-3360L industrial computer [42].

Information sharing with appropriate communication technologies and protocols is one of the key issues in IoT [63]. Efficient IoT communication protocols and technologies are necessary for external air monitoring since the sensors used for the external air detection IoT system are distributed all over the city. The common communication technologies and protocols include ZigBee [41,42,57], GSM [41,54], 2G [53], 3G [44,53,55], 4G [44,45,53,55,57], IEEE 802.11b wireless link [52,60], IEEE 802.15.4 [42,54], WLAN [38,39], WIFI [42,44,52,53,62], Bluetooth [39,51,54,55,58,64], Low Power Wide Area Network [47], Sigfox [47], HTTP [44], Ethernet [53,62], NB-IoT [53], and LoRAWAN [40]. The HTTP protocol ensures compatibility between devices in IoT systems communicating with WIFI technology [44]. Brynda, Kopriva, and Horák [54] combined fixed-position sensors and mobile wearable sensors to monitor air quality and noise pollution. They used Bluetooth to connect fixed-position sensors and mobile wearable sensors. Cao, Yang, Lu, Schultze, Gu, Zhou, Xu, and Lee [58] also used Bluetooth to connect mobile sensors to users’ mobile phones to record the data and obtain position data.

Because external air monitoring IoT systems generate large amounts of data, advanced data analysis methods are necessary. Fog computing [65] and cloud computing [44,45,55,57,63,65,66] are commonly used in data analysis. Usually, fog computing is responsible for data preprocessing, while cloud computing is responsible for data analysis. For example, Wang et al. [65]’s system includes a Local MultiSource Heterogeneous Data Fusion Subsystem (LMFS) and Centralized Homologous Data Training Subsystem (CHTS). The LMFS, in a central cloud, collects data from sensors directly. The CHTS, in distributed fog gateway devices, serves as a distributed federated learning system based on Stochastic Gradient Descent model averaging. The CHTS transfers model parameters for local model training, inference, real-time system, feature engineering, and constructing subclassifiers.

In addition, cloud computing would also be used as a database [44,52,55,57]. Azure, as an example of a cloud pipeline, can store and analyze the sensor-generated data through its applications like Stream Analytics, Azure IoT services, Power BI, and Azure ML [45]. Other scholars may store data in SQL databases, such as MySQL [41,44,62], SQL database [45]. Some of them may store the data in SD cards [38,47].

Data preprocessing is necessary to maintain the accuracy of sensor-generated data. For example, Liu, Chen, Lin, Chen, Chen, Wen, Sun, Juang, and Jiang [41] developed a calibration equation for the sensors through linear regression. Hoang et al. [43] highlighted some data preprocessing techniques, including the measurement of the mean absolute error of the sensor and Limited-Distance-K-means for clustering.

Descriptive analysis and machine learning are major data analysis methods. Descriptive analysis (e.g., line charts [41], bar charts [42]) is used to analyze the data commonly [41,60]. Some scholars may conduct data visualization [43,57]. The methods of data visualization include inverse distance weighted interpolation [43] and bilinear interpolation [46]. For instance, Garau, Nesi, Paoli, Paolucci, and Zamperlin [46] use sensor data to calculate air quality index (AQI) and then construct AQI heatmaps through bilinear interpolation [46]. Wen et al. [42] used segmented regression to transfer the voltage of the sensor to the CO concentration. Multiple linear regression can also be used in data analysis [53]. Currently, machine learning is commonly used to analyze sensor-generated data [44,49,52,55,56,57,63,65,66,67]. The machine learning techniques include neural network [49], support vector machine [52], Long Short-Term Memory [56,67], and Random Forest [44]. There are three challenges in understanding the relationship between environmental parameters and sensor data, including the lack of detailed observation of environmental data in real time, the automated identification of anomalous events from environmental data, and joint multisensor data and multivariate visualization. Therefore, Rathore et al. [57] proposed an effective integrated visualization framework for real-time urban microclimate detection and analysis containing a Bayesian Maximum Entropy-based spatiotemporal identification method and a hyperellipsoidal model-based anomaly detection method. Since atmospheric factors, as well as previous dynamical behavior of its own, may affect the variation in PM diffusion, Lee [49] developed a nonlinear autoregressive network with exogenous inputs from a radial basis function neural network to model the time-dependent causal associations between PM concentration and atmospheric factors. Yu et al. [67] used a Long Short-Term Memory deep learning technique to analyze the weather data from sensors to forecast surface temperature.

After data analysis, people tend to provide a holistic view of sensor-generated information on applications, especially mobile applications [44,49,51,52,55,56,58]. The LabVIEW program is one of the platforms for constructing the applications [41]. Garau et al. [46] introduced the Snap4City platform for people to develop an IoT environmental management platform for smart and sustainable cities using heterogeneous data.

In addition, citizen engagement is one of the research trends in the IoT-based external air quality monitoring system. First, citizens as sensors can play a role in data collection. Huang et al. [55] proposed using crowdsourced automobiles and their built-in sensors to monitor fine-grained air quality in urban environments, which is an effective way to collect high-resolution air quality data anytime, anywhere, and at a relatively low cost. Second, the IoT system should be people-centric. Yu et al. [67] also utilized data collected by sensors deployed on vehicles. They proposed a deep learning network framework based on long and short-term memory to integrate historical in situ observations (i.e., weather station) and IoT observations (i.e., sensors deployed on vehicles) to predict temperatures at high temporal and spatial resolutions. The network comprises two stacked layers of long and short-term memory, which can simultaneously process measurements from all sensor locations and generate predictions for multiple future time steps. Yang et al. [52] proposed a people-centric and cognitive Internet of Things environmental monitoring system to quantify personal exposure to air pollution and evaluated this system in New York. Schürholz et al. [56] developed MyAQI, an IoT air quality monitoring and prediction system that takes into account the context of the external environment and user attributes. This system can send air quality monitoring notifications to users with different sensitivity levels to main pollutants, changing the severity according to user health conditions. Cao et al. [58] developed a mobile temperature and humidity sensor to characterize intracity variations in human thermal conditions and reduce the risk of heat stress. The sensor can connect mobile phones through Bluetooth and allow people to check the surrounding temperature and humidity on mobile applications. Rebeiro-Hargrave et al. [51] introduced MegaSense, a spatially distributed IoT-based monitoring system for urban air quality. This system aims to provide surrounding air information to people.

#### 5.1.2. Noise Monitoring

Noise pollution is a major concern for the public, as it disrupts daily routines and causes discomfort such as headaches, leading to a decline in quality of life [66,68]. As a result, smart cities have begun implementing noise monitoring IoT systems to continuously monitor environmental acoustic parameters at various locations [66]. McDonald et al. [60] introduced an IoT system with delay tolerance units to monitor noise levels in the city center of Dublin. Based on the noise data collected in Dublin, Navarro et al. [68] proposed a big data framework, MapReduce algorithm, to analyze noise data. Mydlarz et al. [62] and Yun et al. [61] both focus on New York’s IoT noise monitoring system, Sounds Of New York City (SONYC). Mydlarz et al. [62] introduced the basic information of SONYC. Yun et al. [61] updated the SONYC by using a real-time Convolutional Neural Network (CNN)-based embedding model, disconnecting from power or network infrastructure support and dynamically choosing good network frequencies.

To monitor urban acoustic environments, people may collect audio data with a microphone [60,61,62,68], remote sonometer (e.g., Model Cesva TA120) [66,69], sound pressure sensors (e.g., MP34DT01) [44]. Yun et al. [61] used TDK InvenSense ICS-43434 as a microphone in the Sounds of New York City IoT system. The processor was an Intel XScale 255 [60], Cypress PSoC4 BLE [64].

Pita et al. [66] used the k-means algorithm to cluster behavior patterns of sound pressure levels geographically and temporally, which is effective for analyzing noise data captured by acoustic sensors in Barcelona. Another research work by Navarro and Pita [69] proved that artificial neural networks can predict long-term noise patterns at a location based on short-term measurements. They also emphasized that increasing the number of hidden layers and collecting an hour of data every hour during the 14:00–22:00 time interval can improve the performance of artificial neural networks.

Since low-cost acoustic pressure sensors are not accurate and measurements are highly variable, Monti et al. [70] introduced RaveGuard, an unmanned noise monitoring platform that utilizes artificial intelligence strategies to improve the accuracy of low-cost devices. They first deployed RaveGuard along with professional sound intensity meters in the center of Bologna, Italy, for more than two months to collect a large amount of accurate noise pollution data. They then applied supervised learning algorithms (with Random Forest achieving the best performance) to train these data into an IoT platform called InspectNoise for use with low-cost microphones to improve the accuracy of their noise detection.

#### 5.1.3. Soil Monitoring

To measure soil health indicators in urban tree pits, Yu et al. [64] proposed Plant Spike, a low-cost and low-power IoT soil monitoring system. The system’s functionality and robustness were demonstrated through testing in New York and Morningside Heights. For soil monitoring, we should collect data including light (e.g., Model OPT3001), temperature, and moisture/capacitive [64]. The model of the light sensor is Model OPT3001 [64].

#### 5.1.4. Integrated Environment Management IoT System

The integrated environment management IoT system is one of the research trends in smart environment management. First, IoT systems can simultaneously monitor indoor and outdoor air quality [51,57,65]. Further, IoT systems are expected to integrate various data sources to offer a comprehensive perspective on the operation of an entire city. For example, Schürholz et al. [56] IoT system combines air quality data, fire incidents, traffic volume, and user data (geolocation, user ID, pollutant sensitivity, timestamps). Monti et al. [70]’s RaveGuard platform consists of a noise pollution sensing system (i.e., a USB condenser microphone mounted on a Raspberry Pi 2 Model B) and an environmental sensing system (i.e., a Canarin II system, which can detect temperature, relative humidity, barometric pressure, and particulate matter, specifically PM 1. 0, 2.5, and 10).

### 5.2. Smart Waste Management

Efficient waste management can prevent environmental problems and reduce costs [71,72]. Smart bins are the major research area in smart waste management, since they can reduce waste and improve the efficiency of waste management. For example, Hong et al. [71] implemented an IoT-based smart garbage system in Seoul, which reduced the average amount of food waste by 33% over the course of a year.

In general, the operation of smart bins includes recognizing a passerby and then opening the lid for them, using sensor weighing to detect bin capacity status, planning the best path for the garbage collection truck, and allowing citizens to view garbage can statuses in applications. Hong et al. [71] used an RFID reader to identify the passerby and open the lid of the smart bin. In general, the smart bin is weighed by an ultrasonic sensor and will call the truck to collect the garbage when it reaches a certain weight [71,73,74]. Ultrasonic sensors usually determine the level of garbage by measuring the time interval between sending sound and receiving echo. Ultrasonic sensors are usually placed at the bottom of the smart bin to measure weight [71]. The ESP8266 NodeMCU is an example of a processor that can be used to install sensors. [74]. Waste sorting at the source can achieve efficient waste management and improved urban sustainability [75,76]. Aloui et al. [74] detect metal with an inductive sensor. Their smart bin also collects data on shocks and vibrations. All the smart bins scattered around the city will be connected together through some IoT communication protocols/technology. For example, Hong et al. [71] used CDMA2000 EV-DO to connect smart bins. Then, we collected smart bin location data using GPS to plan the path of the garbage collection truck. For example, Aloui et al. [74] utilized the GPS NEO-6M model. The methods for vehicle route planning include Tabu search [73], Ant Colony [73], and Genetic Algorithm [73]. Sarvari et al. [73] suggested considering sensor-generated fill levels of bins at customer premises, varying service durations depending on the customer, the time availability of each truck driver, the capacity of each truck, the availability of each customer, and the prioritization of collection services based on each customer’s needs in collection vehicle problem planning. They found that Genetic Algorithms are better suited to solving time-windowed waste collection vehicle routing problems. Finally, the smart bin may have an application to provide information to users and administrators [71]. For example, Aloui et al. [74] proposed an IoT waste management system that integrates all phases, considering user attributes, health and environmental standards, and other factors. This system allows for the selection of the optimal route, real-time monitoring of container data, tracking of trucks, waste categorization, identification of peak waste time and risks, and generation of waste management statistics and reports.

## 6. Smart Transportation

SDG 11.2 indicates that “By 2030, provide access to safe, affordable, accessible and sustainable transport systems for all, improving road safety, notably by expanding public transport, with special attention to the needs of those in vulnerable situations, women, children, persons with disabilities and older persons” (https://sdgs.un.org/goals/goal11#targets_and_indicators, accessed on 10 February 2024). IoT can enhance the sustainable development of smart transportation by improving traffic efficiency, parking lot utilization, and public transportation efficiency.

### 6.1. The Improvements of Traffic Efficiency

Previous literature has discussed methods for enhancing transportation efficiency, primarily through the efficient processing of data from smart transportation IoT, citizen engagement, and the use of IoT-enabled transportation infrastructure.

In smart transportation IoT systems, scholars generally use presence sensors (e.g., GPS [77], Bluetooth sensor [78], and infrared laser light sensor [79]) to detect passing people or vehicles and speed sensors (e.g., inductive loop detector [77], optical/Laser sensors [80]) to detect their speed. Cellular data transmission (e.g., GSM [77], 3G [79]) is the most commonly used technology for data transmission.

Scholars have different insights on efficiently processing data from smart transportation IoT systems. Saravanan and Kumar [77] developed an IoT traffic system. This system can predict traffic congestion by comparing normalized speed and identify traffic accidents through the accident detection model. Rathore et al. [81] suggested to use parallel processing systems and big graph processing technology to process big data from IoT transportation systems. De Iasio et al. [82] used IoT technology, cloud computing, fog computing, microservices, and DevOps infrastructure to develop the PROMENADE platform for real-time monitoring and analysis of transportation data generated by IoT devices in large smart cities. This platform combines cloud computing, fog computing, and IoT technology. It can provide information such as shortest paths and the betweenness centrality of each node. Bandaragoda et al. [78] proposed a framework for real-time smart commuting behavior analysis based on artificial intelligence and IoT. Traffic data are obtained in real time via Bluetooth sensors located on Victoria’s main roads. When a vehicle with a Bluetooth device passes through its scanning area, the MAC address of the Bluetooth device is recorded by the scanner along with the timestamp of that event and transmitted to a central database. The data generated by the sensor include the analysis of traffic flow and traffic trajectory under stable and fluctuating commuting behavior scenarios. To analyze pedestrian mobility, Carter et al. [79] used infrared laser light sensors to scan and count passing pedestrians. They conducted static analysis using Microsoft Excel^®^ and more advanced interactive visualizations and analyses with PowerBI^®^.

Citizen engagement plays a crucial role in smart transportation IoT systems, as citizens are significant participants in transportation systems. Therefore, the data from the smart transportation IoT system can be analyzed in conjunction with citizen participation data. Multiple modal data can complement and reinforce each other [83]. For example, Fonseca et al. [80] combine the quantitative transportation data and the qualitative transportation data from citizen engagement to obtain more accurate data. Social media can be a useful way to access citizen engagement data. Chen and Wang [84] and Chen et al. [83] both combined sensor-generated data and social media data for traffic event detection. Chen and Wang [84] suggested using a Multimodal Neural Network. They used a Recurrent Neural Network to process sensor-generated data and a Convolutional Neural Network to process social media data. Chen et al. [83] suggested using a multimodal generative adversarial network. They used LSTM-RNN for the feature extraction of sensor-generated data in this network.

IoT-enabled transportation infrastructures include IoT road monitoring systems and IoT lampposts. Ho et al. [85] introduced a real-time IoT-based system to monitor the occupancy of loading and unloading bays on roadsides in Hong Kong. This system used high-definition smart cameras to collect data, including vehicle parking space regulations, types of vehicles, and the minimum width of traffic lanes. The system transmits data with wireless communication. Based on data analysis on the data from cameras, the system can evaluate parking gaps and provide decision support on parking spaces. Cacciatore et al. [86] proposed an IoT LED lamppost for street lighting in Luxembourg City. These lampposts use presence sensors (e.g., SE-10 PIR motion sensor) to detect the presence of citizens nearby and dim light intensity to achieve lower energy and cost.

### 6.2. The Optimization of Parking Lot Utilization

To enhance parking lot usage, scholars tend to develop an IoT-based application to provide parking availability information and guide parking routes. Doukas et al. [87] introduced COMPOSE, an open-source platform as a service, to develop the IoT transportation systems in Barcelona. This system collects the data of users’ position by GPS and the data of parking with parking sensors in the city. Pham et al. [88] developed a GS1 smart parking system to deal with the sensor-generated data in the parking scenario. This system consists of EPC Information Service, Object Name Service, and Area-2-GLNs mapping service. This system has been used in Busan City and nine Korean airports. Sotres et al. [89] presented a global smart parking platform covering the IoT smart parking systems in five smart cities, including Barcelona, Santander, Seoul, Busan, and Seongnam. There are two types of parking sensors used in these cities. One type of parking sensor is based on ferromagnetic detection, which is buried under the asphalt. They can record the percentage of time a vehicle is on top of the magnetic loop. These sensors also can record the total counted vehicles per hour and a traffic congestion index, as well as model traffic congestion. Another type of parking sensor is based on radar sensing, which is mounted over the floor. The sensors communicate the data through a low-power wide-area network.

Regarding data analysis for IoT smart parking systems, Zheng et al. [90] found that a regression tree is the least computationally intensive algorithm that performs the best for parking availability prediction in the real-time car parking datasets of San Francisco City and Melbourne City. Zhu and Yu [91] used a three-axis AMR sensor to generate magnetic signals and applied normalized cross-correlation (i.e., k-means clustering algorithm) as their data analysis method, improving the detection accuracy for on-street parking. Atif et al. [92] used a discrete Markov chain model to propose the SmartPark algorithm to analyze the data of an IoT parking system and reveal the future state of the parking lot when vehicles are expected to arrive at the parking lot.

### 6.3. The Improvements of Public Transportation Efficiency

IoT systems can improve the efficiency of subways, cabs, and shared bikes.

Regarding subways, Lee et al. [93] introduced an IoT-based train management system that improves the operational efficiency of the railway in Hong Kong. This system can conduct a holistic analysis of the data from multiple sensors, such as the sensors monitoring the temperature of the rolling stock axle counter, rail integrity, and equipment rooms, GPS and RFID monitoring train speed and position, and V-sensor monitoring the vibrations of the bogie. After data analysis, the system will distribute the tasks to the subsystems. For instance, the system may trigger the train’s public address system to broadcast passenger information upon arriving or departing a station. Li et al. [94] proposed a two-stage adaptive model for forecasting passenger flows on large-scale rail systems. The first stage introduces a self-focused prediction-based model that predicts the next day’s traffic based on historical data from IoT devices, and the second stage develops a real-time fine-tuning model that adjusts the predicted traffic based on emergencies and short-term traffic variations from physical network devices. They used load sensors to collect data on passenger flow.

For cabs, optimal rebalancing allows traffic to move passengers in uneven directions with a minimum number of vehicles. However, long waits for cab rides during peak hours can reward quality of service. Therefore, Chuah et al. [95] limited the waiting time in the passenger queue to a specified range in optimal rebalancing, which extends the autonomous on-demand mobility solution for fleets of connected autonomous cabs.

Public bike-sharing systems are the most popular public transportation systems in the world [96]. Alaoui and Tekouabou [97] proposed IoT and machine learning techniques to optimize the management of autonomous bike-sharing systems in smart cities through a regression integration approach. They found that the Random Forest Regressor, Bagging Regressor, and XGBoost Regressor are more suitable for analyzing the data in the bike-sharing system. The large number of e-fences caused high difficulty and cost in scheduling the shred bikes [98]. Wu et al. [98] proposed a process planning algorithm for electric dispatching vehicles to schedule the shared bikes. This algorithm includes demand prediction (i.e., LSTM-GRU) and scheduling subarea division.

### 6.4. Accessibility

Kamaldin et al. [99] introduced SmartBFA, an IoT system that provides peer-to-peer accessible information for persons with disabilities. They deployed GPS and an inertial measurement unit on electric wheelchairs. Prandi et al. [100] developed an IoT system that provides indoor wayfinding functionality to improve indoor mobility. They used smart beacons as sensors to locate people.

## 7. Disaster Management

SDG 11.5 indicates that “By 2030, significantly reduce the number of deaths and the number of people affected and substantially decrease the direct economic losses relative to global gross domestic product caused by disasters, including water-related disasters, with a focus on protecting the poor and people in vulnerable situations” (https://sdgs.un.org/goals/goal11#targets_and_indicators, accessed on 10 February 2024). The primary function of the IoT in urban disaster management is predisaster prediction.

In predisaster monitoring, people use IoT systems for structural health monitoring and crowd monitoring. Bennett et al. [101] discussed two IoT structural monitoring systems in the Prague Metro and London Underground. They used crackmeters (i.e., linear potentiometric displacement transducers), inclinometers (Model: SCA103T-D04), and humidity and temperature sensors (Model: Sensirion SHT11). They used the Crossbow MIB600 (XMesh routing protocol) as a gateway to collect the sensor-generated data and then transmit them to a single computer board through an Ethernet cable. The single computer board then transmits the data to a database through GPRS. Ni et al. [102] deployed an IoT-based structural health monitoring system for the new headquarters of the Shenzhen Stock Exchange in Shenzhen City. Since this building is one of the largest cantilever structures worldwide, it is important to measure the strain and deflection of the cantilever structure. The sensors in the system included a vibrating-wire strain gauge, temperature sensor, and accelerometer. The system also had a vision-based displacement tracking system. The communication technology is WLAN. Shen et al. [103] used a vibrating-wire strain and temperature sensor, a three-axis digital output linear accelerometer (i.e., Model LIS3LV02DQ), a wind speed sensor, and a displacement sensor to develop an IoT-based structural health monitoring system for the National Stadium in China. The vibrating-wire strain and temperature sensor integrated a digital temperature chip (i.e., Model DS18B20). They used IEEE 802.15.4/ZigBee for communication. Huang et al. [104] proposed a structural health monitoring IoT system to monitor section convergence, joint tensioning, and water seepage in the segmental lining of shield tunnels with light-emitting diodes to indicate different risk levels for workers on site, and finally, an elasticity-based analytical model for designing rehabilitation strategies. Their IoT system included a tilt sensor, crack sensor, and seepage sensor. They used Zigbee and WiFi as communication protocols. Khemapech [105] deployed an IoT-based structural health monitoring system in at a concrete bridge in Bangkok City. The system is on a Microstrain platform. This system used an applied strain gauge as a sensor to measure bending strain. The communication protocol is Lossless Extended Range Synchronized. They adopted real-time stream processing and artificial neural network techniques in data analysis. For an excepted structural health monitoring system, Domingo [106] used a deep learning algorithm to process crowd data and predict the number of people at Castelldefels Beach in Barcelona City, which is important for rescue operations at the beach. The crowd data are generated by cameras. They used “CountThings” as image recognition software. In addition, Xu et al. [107] proposed a blockchain-based IoT system for structural health that provides verification of monitoring privileges, generates anomaly alerts, ensures data invariance, resists attacks, and allows for traceability queries.

## 8. Emerging Applications

The IoT can also contribute to the sustainable development of smart cities in terms of smart agriculture, smart education, smart tourism, and smart healthcare.

In addition, Qiu et al. [108] and Guo et al. [109] introduced IoT systems for smart agriculture. Qiu et al. [108] applied an IoT system to monitor and analyze the facility agricultural ecosystem in Shanghai. Guo and Zhong [109] applied an IoT system for span greenhouse agricultural system in Tianjin. Their system included the automatic control function of the greenhouse and the agricultural intelligent frequency conversion irrigation function.

For smart education, Yang and Hsu [110] installed the ESPSAS IoT platform in an elementary school in Taipei. ESPSAS allowed students to create their own windflower things, watch their own windflower spinning on campus via mobile phones, and change its light color remotely via mobile phones.

Regarding smart tourism, Davoli et al. [111] utilized an IoT system with ultrasonic sensors to help an XR tourism application sense the user’s surroundings. 

For smart healthcare, Liu et al. [112] introduced eBPlatform, an IoT system deployed in a patient’s home that can measure the patient’s blood glucose, blood pressure and ECG signals. In addition, Salehi-Amiri et al. [113] used IoT to address the uncertain home healthcare supply chain network. Using data from a real-world case study in Montreal, they found that leveraging this IoT system could maximize the overall utilization of vehicles and efficiently optimize the number of trips, thereby minimizing total costs and greenhouse gas emissions.

## 9. Privacy and Security

Two publications highlighted the privacy and security of IoT applications in smart cities. The Snap4City solution proposed by Badii et al. [114] can address all the security issues in the whole IoT system. Javed et al. [115] designed a security module pluggable with O-MI nodes in their IoT system to ensure access control, data confidentiality, integrity, and data encryption.

## 10. Future Research Directions

Future research should focus on the integration between IoT systems for indoor environmental management and smart home devices (e.g., smart windows, smart air purifiers, smart temperature control systems). In other words, IoT systems can automatically adjust the indoor environment based on environmental data and automatically provide a comfortable living environment for humans, which is especially useful for people with asthma, allergies, and other diseases.

Regarding smart waste management, one future research direction is that smart bins have the ability to identify toxic metal waste, such as batteries. In addition, we can use cameras and computer vision algorithms to identify each type of waste, providing a more refined waste classification.

For smart transportation, IoT systems play an important role in connected and autonomous vehicles, shared mobility systems (SMSs), and mobility-as-a-service (MaaS). SMSs are articulated transportation systems based on digital infrastructure that enable flexible sharing of vehicles and routes, helping to optimize resource utilization and facilitate collaboration across the mobility chain [116]. The IoT can connect autonomous vehicles and construct SMSs. Each vehicle can have GPS sensors to upload its position data to the central cloud, which can optimize the route of the vehicles in the whole system. Other transportation modes, such as bicycles, buses, and ships, can also be added to the SMS, which can be further developed into MaaS. MaaS integrates multiple transportation modes into one service platform [117]. Therefore, the ultimate MaaS platform can combine different modes of transportation to help users plan the fastest route to their destination. By analyzing global data, MaaS can organize autonomous vehicles to pick up different passengers on efficient routes, increasing operational efficiency and reducing carbon emissions on the way to pick up passengers.

Furthermore, it has been discovered that citizen engagement is a crucial factor in the development of smart communities, smart transportation, and disaster management. Therefore, we suggest that future research combines IoT and citizen engagement. Social media can be a useful way to access citizen engagement data.

Last but not least, one of the important future research directions is that an IoT system can cover most aspects of smart cities, because some sensors, communication technologies and protocols, and data analyses and applications are universal. This is important because it can reduce costs and remain environmentally friendly. Figure 3 describes the framework of this universal IoT system, which covers most aspects of smart cities. Citizens can use this integrated IoT system through a mobile application.

## 11. Conclusions

This study conducted an SLR to investigate the role of the IoT in the sustainable development of smart cities. The analysis of 73 publications revealed that IoT plays a crucial role in the sustainable development of smart cities, especially in the areas of smart communities, smart transportation, and disaster management. A smart community combines smart environmental management and smart waste management [6]. Smart environment management includes external and internal air quality monitoring, urban noise monitoring, and soil monitoring. Smart bins are the main area in smart waste management. Smart transportation includes traffic management, smart parking, and public transportation. These areas are interconnected. For instance, IoT systems for intelligent transportation enhance transportation efficiency, resulting in reduced vehicle emissions, improved energy efficiency, and decreased environmental pollution. However, it is important to note that this study has limitations. For instance, the use of keywords may result in the omission of some literature in this research area, despite being formulated based on the research questions. Additionally, there are emerging areas that require further investigation, including smart agriculture, smart education, smart tourism, and smart healthcare.

The findings in our study extend the studies of Ramírez-Moreno et al. [14] and Bibri and Krogstie [118]. Compared with the study of Ramírez-Moreno et al. [14], our study highlights the role of IoT systems in environment monitoring and disaster management. Compared with the study of Bibri and Krogstie [118], our study focuses on IoT systems and summarizes the role of IoT systems in the sustainable development of smart cities by reviewing academic studies. What is more, our study tends to develop a framework to integrate the IoT systems in different aspects of sustainable development into one mobile application, which can reduce costs and energy consumption and facilitate use by citizens.

## Figures and Tables

**Figure 1 sensors-24-02074-f001:**
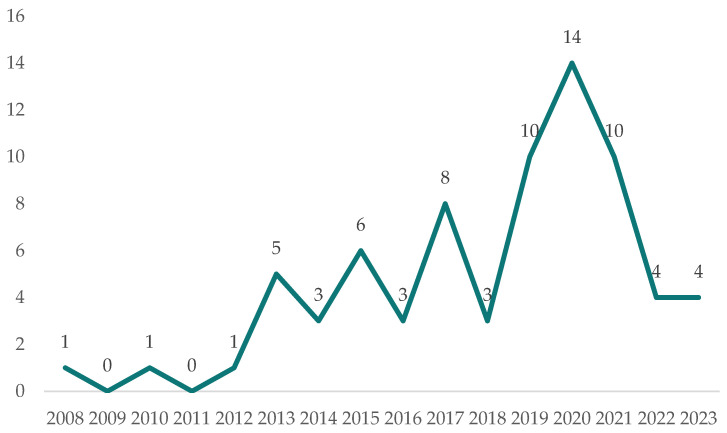
The distribution of publications by year.

**Figure 2 sensors-24-02074-f002:**
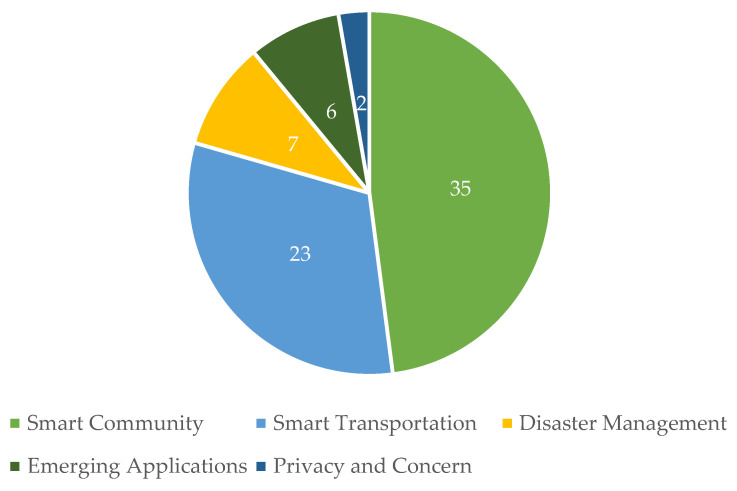
The distribution of publications by topics.

**Figure 3 sensors-24-02074-f003:**
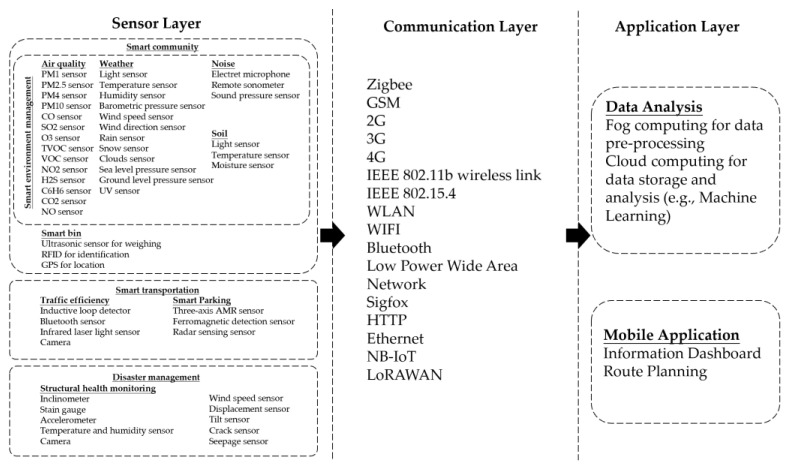
IoT system that covers most aspects of smart cities.

**Table 1 sensors-24-02074-t001:** Top six distribution of publication by sources.

Source	Number of Publications
Sensors	7
IEEE Internet of Things Journal	5
IEEE Access	4
Applied Sciences-Basel	2
International Journal of Distributed Sensor Networks	2
International Journal of Environmental Research and Public Health	2
Expert Systems with Applications	2

**Table 2 sensors-24-02074-t002:** The parameters collected for air quality data.

Parameter	Model	Reference
PM_1_	SPS30, Sensirion, Stäfa, Switzerland	[45,51]
PM_2.5_	PPD-60V2	[52]
	PMS5003, Plantower, Nanchang, China	
	PMS5003-G5	[44]
	SPS30, Sensirion AG, Stäfa, Switzerland	[51]
	Not specified	[44,45,46,48,49,51,52,53,54,55,56]
PM_4_	SPS30, Sensirion AG, Stäfa, Switzerland	[51]
PM_10_	SPS30, Sensirion AG, Stäfa, Switzerland	[45,51]
	Not specified	[45,46,49,51,53,54,56]
Carbon monoxide (CO)	CO-B4, Alphasense, Essex, UK	[45]
	MiCS-4514	[51]
	MiCS-5525 metal oxide semiconductor	[41,42]
	TGS 2442	[43]
	A3CO, City Technology Ltd., Portsmouth, UK	[50]
	Not specified	[41,42,43,46,50,52,53,54,56]
SO_2_	SO_2_-B4, Alphasense, Essex, UK	[45]
	Not specified	[46,52,53,54,56]
O_3_	MQ-131	[51]
	OX-B431, Alphasense, Essex, UK	[45]
	Not specified	[46,51,52,53,56]
TVOC	Not specified	[55]
VOC	Not specified	[54]
NO_2_	MiCS-4514	[51]
	NO_2_-B43F, Alphasense, Essex, UK	[45]
	Not specified	[46,51,53,54,56]
H_2_S	Not specified	[46]
C_6_H_6_	Not specified	[46]
CO_2_	SenseCAP AU915	[40]
	Aranet 4 Pro, Aranet, Riga, Latvia	[40]
	Not specified	[40,46]
NO	Not specified	[46]

**Table 3 sensors-24-02074-t003:** The parameters collected for weather data.

Parameter	Model	Reference
Light	S1087, HAMAMATSU PHOTONICS K.K., Shizuoka, Japan	[41]
	OPT3001	[44]
	Not specified	[41,44,57]
Temperature	WS-2316, La Crosse Technology, La Crosse, WI, USA	[41]
	BME280	[51]
	SHT11, Sensirion AG, Stäfa, Switzerland	[41,44,45]
	SHTC3, Sensirion AG, Stäfa, Switzerland,	[41,44,45]
	Si7021-A20-GM1	[41,44,45]
	Model TMP007	[44]
	Not specified	[37,40,41,46,51,53,54,55,56,57,58]
Humidity	WS-2316, La Crosse Technology, La Crosse, WI, USA	[41]
	BME280	[51]
	SHT11, Sensirion AG, Stäfa, Switzerland	[41,44,45]
	SHTC3, Sensirion AG, Stäfa, Switzerland,	[41,44,45]
	Si7021-A20-GM1	[41,44,45]
	Not specified	[37,40,41,46,51,53,54,55,56,57,58]
Barometric pressure	WS-2316, La Crosse Technology, La Crosse, WI, USA	[41]
	BMP280	[44,51]
	BMP390L	[45]
	Not specified	[40,41,46,51,53,54,58]
Wind speed	WS-2316, La Crosse Technology, La Crosse, WI, USA	[41]
	Not specified	[41,46,54,56]
Wind direction	WS-2316, La Crosse Technology, La Crosse, WI, USA	[41,56]
	Not specified	[41,46,54]
Rain	WS-2316, La Crosse Technology, La Crosse, WI, USA	[41]
	Not specified	[41,46]
Snow	Not specified	[46]
Clouds	Not specified	[46]
Weather description	Not specified	[46]
Sea level pressure	Not specified	[46]
Sunrise and sunset	Not specified	[46]
Ground level pressure	Not specified	[46]
UV	SI1133-AA00-GM	[51]

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
