# Peer review of "Sensors on Internet of Things Systems for the Sustainable Development of Smart Cities: A Systematic Literature Review"

_sensors, 2024, doi:10.3390/s24072074_

Round 1
Reviewer 1 Report
Comments and Suggestions for Authors
In this review paper, sensors in internet of things systems for the sustainable development of smart cities, the literature is systematically reviewed and the results are shared. In general, the organization and presentation of the paper is appropriate. Research questions for the systemic review study are listed and the main objective is clearly stated. In the methodology section, keywords, inclusion, and exclusion criteria were determined and applied. However, Sensor in Smart city applications is a very limited topic and is quite narrow in scope for the literature review study. The study addresses a limited audience of readers and researchers, and detailed evaluations of the paper are listed below.
1) While the sensor types to be taken into consideration are included in IoT systems, less attention is given to smart city applications.
2) In the paper, I could not see any explanation about the study selection process. This should be explained in a few sentences. This requires a clearly defined study selection process, including screening of titles, abstracts, and full texts.
3) A systematic and repeatable approach should be used to ensure transparency. Additionally, the information to be extracted from each included study should be stated. This may include details about sensor types, applications, sustainability measures, challenges, and outcomes.
4) Describe how the data from included studies will be synthesized and analyzed. This may involve thematic analysis, meta-analysis, or other relevant methods.
5) Please provide a critical appraisal of the findings from the included studies. Discuss any patterns, trends, or discrepancies in the literature.
6) Heterogeneity among included studies, including differences in methodologies, populations, and settings, should be acknowledged, and addressed. State clearly how this heterogeneity was addressed in the review.
7) Paper also does not have a coherent structuring of the topic (such as methodological approaches, chronological order)
8) Listed below are recent review papers with similar content. What are the different aspects of the present study from these studies?
a. Ramírez-Moreno, M. A., Keshtkar, S., Padilla-Reyes, D. A., Ramos-López, E., García-Martínez, M., Hernández-Luna, M. C., ... & Lozoya-Santos, J. D. J. (2021). Sensors for sustainable smart cities: A review. Applied Sciences, 11(17), 8198.
b. Bibri, S. E. (2018). The IoT for smart sustainable cities of the future: An analytical framework for sensor-based big data applications for environmental sustainability. Sustainable cities and society, 38, 230-253.
Author Response
1) While the sensor types to be taken into consideration are included in IoT systems, less attention is given to smart city applications.
→ Thank you for your comments. We discuss about smart city applications in terms of smart community, smart transportation, disaster management and emerging applications. We discussed about the application of IoT in smart community in the line 248-488. We talked about the application of IoT in smart transportation in the line 489-590. We talked about the application of IoT in disaster management in the line 591-634. And there are some emerging applications. Please kindly find them on the line 635-654.
2) In the paper, I could not see any explanation about the study selection process. This should be explained in a few sentences. This requires a clearly defined study selection process, including screening of titles, abstracts, and full texts.
→ Thank you for your comments. The selection process consists of two stages. During the first stage, publications related to the application of IoT in the sustainable development of smart cities are selected through screening of their titles, abstracts, and keywords. In the second stage, the full text of each selected publication is read to ensure it falls within the scope of our focus. Furthermore, we only consider the studies connecting to the real world, which can accelerate the application of the research findings. That is, the scholars may apply their research outcome to smart cities or use sensor-generated data in smart cities for data analysis. Please kindly find them on the line 191-199.
3) A systematic and repeatable approach should be used to ensure transparency. Additionally, the information to be extracted from each included study should be stated. This may include details about sensor types, applications, sustainability measures, challenges, and outcomes.
→ Thank you for your comments. We record the information on Appendix B. Please kindly find them on the line 728.
4) Describe how the data from included studies will be synthesized and analyzed. This may involve thematic analysis, meta-analysis, or other relevant methods.
→ Thank you for your comments.
Firstly, we conduct descriptive analysis for 75 publications to present the trend of “the sustainable development of smart cities”. Descriptive analysis includes the publication number distribution by year, and by source.
Secondly, to figure out the major topics related to the sustainable development of smart cities, we may conduct thematic analysis through reading the abstracts of the publications. The categorization will be also revised while reading the full text of the publication. For thematic analysis, researchers usually categorize the data into theme based on recurring ideas.
Thirdly, we conduct content analysis for each included publication. We systematically analyze the full text of each publication to identify the meanings within the content. we summarize key findings and record the basic information on appendix B. Appendix B presents each included publication’s information such as sensor type, application, sustainability measure, challenge, and outcome. We synthesize the findings to identify the pattern and trend. For the sample publications in same topic, we discuss them in chronological order so as to facilitate understanding of the technology's development. Then, we discuss discrepancies and inconsistencies and write a critical appraisal of the findings. We also explain the heterogeneity among included publications, such as differences in methodologies, environment, and the construct of IoT systems.
Finally, we conduct framework analysis to integrate the IoT systems for different aspects of sustainable development of smart cities.
Please kind find them on the line 200-219.
5) Please provide a critical appraisal of the findings from the included studies. Discuss any patterns, trends, or discrepancies in the literature.
→ Thank you for your comments. We discuss discrepancies and inconsistencies and write a critical appraisal of the findings. Please kindly find them on the line 214-215.
6) Heterogeneity among included studies, including differences in methodologies, populations, and settings, should be acknowledged, and addressed. State clearly how this heterogeneity was addressed in the review.
→ Thank you for your comments. We also explain the heterogeneity among included publications, such as differences in methodologies, environment, and the construct of IoT systems. Please kindly find them on the line 215-217.
7) Paper also does not have a coherent structuring of the topic (such as methodological approaches, chronological order)
→ Thank you for your comments. For the sample publications in same topic, we discuss them in chronological order so as to facilitate understanding of the technology's development. Please kindly find them on the line 213-214.
8) Listed below are recent review papers with similar content. What are the different aspects of the present study from these studies?
- Ramírez-Moreno, M. A., Keshtkar, S., Padilla-Reyes, D. A., Ramos-López, E., García-Martínez, M., Hernández-Luna, M. C., ... & Lozoya-Santos, J. D. J. (2021). Sensors for sustainable smart cities: A review. Applied Sciences, 11(17), 8198.
- Bibri, S. E. (2018). The IoT for smart sustainable cities of the future: An analytical framework for sensor-based big data applications for environmental sustainability. Sustainable cities and society, 38, 230-253.
→ Thank you for your comments. The findings in our study extend the studies of Ramírez-Moreno, Keshtkar, Padilla-Reyes, Ramos-López, García-Martínez, Hernández-Luna, Mogro, Mahlknecht, Huertas, Peimbert-García, Ramírez-Mendoza, Mangini, Roccotelli, Pérez-Henríquez, Mukhopadhyay and Lozoya-Santos [14] and Bibri and Krogstie [113]. Compared to the study of Ramírez-Moreno, Keshtkar, Padilla-Reyes, Ramos-López, García-Martínez, Hernández-Luna, Mogro, Mahlknecht, Huertas, Peimbert-García, Ramírez-Mendoza, Mangini, Roccotelli, Pérez-Henríquez, Mukhopadhyay and Lozoya-Santos [14], our study highlights that the role of IoT systems in the environment monitoring and disaster management. Compared to the study of Bibri and Krogstie [113], our study focuses on the IoT system and summarizes the role of IoT systems in the sustainable development of smart cities by reviewing the academic studies. What is more, our study tends to de-velop a framework to integrate the IoT systems in different aspects of sustainable development into one mobile application, which can reduce the cost and energy con-sumption, and facilitate the use by citizens. Please kindly find them on the line 705-718.
Reviewer 2 Report
Comments and Suggestions for Authors
Authors present an extensive review on sensors, technologies, protocolos, and applications of IoT systems in Smart cities. The paper is very well structured and results interesting to read and analyze.
My only concern is about the search string that was defined for the review. It contains an extensive final part with a listing of cities, where at least one of them must be included. How different were search results without this part of the search string? I understand it was to include the cities from a top 100 list, but some interesting studies from cities not on this list might have been omitted. Please elaborate on this part.
I also think the final sentence of the introduction section may be omitted. The one in lines 80-82.
In general, the document is easy to read. However, I recommend to check the writing style in the Conclusions section.
References 22, 29, 37, 51, 59, 74, and 75 include the DOI two times.
Some references include the DOI as a full URL (doi:http://doi.org./DOI) while other are in the form doi:DOI.
Final paragraph in section 6 has a larger font size (lines 499-504).
Comments on the Quality of English LanguageThere are a few corrections needed:
- Line 288. Remove "to" before the word "developed".
- Line 352. Remove the letter 't' at the end of "preprocessingt".
- Line 394. There are two choices here: "... a smart bin includes sensing ..." if the listing that follows are all the possibilities; "... a smart bin can include sensing ..." if they are a few examples. Also, add a comma after "weighing".
- Line 400. "weighing sensor".
Author Response
Authors present an extensive review on sensors, technologies, protocolos, and applications of IoT systems in Smart cities. The paper is very well structured and results interesting to read and analyze.
My only concern is about the search string that was defined for the review. It contains an extensive final part with a listing of cities, where at least one of them must be included. How different were search results without this part of the search string? I understand it was to include the cities from a top 100 list, but some interesting studies from cities not on this list might have been omitted. Please elaborate on this part.
→ Thank you for your comments. We only consider the studies connecting to the real world, which can accelerate the application of the research findings. That is, the scholars may apply their research outcome to smart cities or use sensor-generated data in smart cities for data analysis. This may be a limitation. We believe that it would be one of the future research directions.
I also think the final sentence of the introduction section may be omitted. The one in lines 80-82.
→ Thank you for your comments. We have removed the sentence in lines 80-82.
In general, the document is easy to read. However, I recommend to check the writing style in the Conclusions section.
References 22, 29, 37, 51, 59, 74, and 75 include the DOI two times.
Some references include the DOI as a full URL (doi:http://doi.org./DOI) while other are in the form doi:DOI.
Final paragraph in section 6 has a larger font size (lines 499-504).
→ Thank you for your comments. We have modified the font size in section 6 (It is section 11 now, lines 693-718).
Comments on the Quality of English Language
There are a few corrections needed:
- Line 288. Remove "to" before the word "developed".
→ Thank you for your comments. We have removed the word “to”.
- Line 352. Remove the letter 't' at the end of "preprocessingt".
→ Thank you for your comments. We have removed the word “t”.
- Line 394. There are two choices here: "... a smart bin includes sensing ..." if the listing that follows are all the possibilities; "... a smart bin can include sensing ..." if they are a few examples. Also, add a comma after "weighing".
→ Thank you for your comments. We have removed the word “can” and added a comma after “weighing”.
- Line 400. "weighing sensor".
→ Thank you for your comments. We have changed the word “weighting” to “weighing”.
Reviewer 3 Report
Comments and Suggestions for Authors
This paper presents the authors’ review on the use of Internet of Things (IoT) in smart cities, and a systematic literature review method was used.
This paper is mainly descriptive. In this paper, the authors described the key components of IoT, including all kinds of sensors and protocols used and the type of information they can collect, as well as its application in many areas, such as environment management, waste management, and transportation. The authors did a good job in summarising, and it is a highlight of this paper.
However, in a review paper, there should be another part, i.e., the authors’ own insights into the problem they are investigating, besides the descriptive description, for further pointing out the directions for future research. This critical part is right missing from this paper.
In addition, when introducing new concepts, give a brief introduction first, such as ‘Santander City’ and ‘Nice City’.
Author Response
This paper presents the authors’ review on the use of Internet of Things (IoT) in smart cities, and a systematic literature review method was used.
This paper is mainly descriptive. In this paper, the authors described the key components of IoT, including all kinds of sensors and protocols used and the type of information they can collect, as well as its application in many areas, such as environment management, waste management, and transportation. The authors did a good job in summarising, and it is a highlight of this paper.
However, in a review paper, there should be another part, i.e., the authors’ own insights into the problem they are investigating, besides the descriptive description, for further pointing out the directions for future research. This critical part is right missing from this paper.
→ Thank you for your comment. We make a new section to discuss future research directions. Please kindly find them on the line 660-691.
In addition, when introducing new concepts, give a brief introduction first, such as ‘Santander City’ and ‘Nice City’.
→ Thank you for your comment. They are the city name. We made a table to present top 100 smart cities in the IMD Smart City Index 2023 list on Appendix A. Please kindly find them on the line 725.
Reviewer 4 Report
Comments and Suggestions for Authors
The main contribution of this paper is to provide valuable insights into the utilization of IoT in the context of smart cities, with a particular focus on its implications for sustainable urban development. The paper aims to bridge the knowledge gap and address the scarcity of comprehensive understanding regarding the role of IoT in fostering sustainable development within smart cities.
I have some comments to enhance the scientific depth of the paper:
1. In the Methodology section, please explain how the data were collected, including the specific information extracted (e.g., attributes of IoT sensors, communication technologies, data analysis methods). Moreover, the authors should define the inclusion and exclusion criteria used to select the publications for the review.
2. In the descriptive analysis section, please provide more analysis to cover a wide range of relevant publications.
3: The authors should discuss how IoT technologies contribute to improving various aspects of community life, such as transportation, energy management, public safety, healthcare, and social services. Moreover, they should give more explanations of the positive impacts on sustainability, resource utilization, and citizen engagement.
4: The authors should provide more discussions on the potential risks and vulnerabilities and propose strategies to ensure data privacy, security, and ethical use of IoT technologies.
5: More discussion is still needed to describe how IoT technologies contribute to improving transportation efficiency, reducing congestion, enhancing safety, and promoting sustainability.
6: The authors should provide a detailed explanation of how ITS integrates various IoT technologies, such as sensors, communication networks, and data analytics, to enable real-time monitoring, control, and management of transportation systems.
7: The authors also need to discuss the potential areas of innovation and research that can further enhance the role of IoT in transportation.
8: What about the integration of emerging technologies, such as connected and autonomous vehicles, shared mobility, and mobility-as-a-service (MaaS), and their potential impact on smart transportation?
Comments on the Quality of English LanguageMinor editing of English language required
Author Response
- In the Methodology section, please explain how the data were collected, including the specific information extracted (e.g., attributes of IoT sensors, communication technologies, data analysis methods). Moreover, the authors should define the inclusion and exclusion criteria used to select the publications for the review.
→ Thank you for your comments. The selection process consists of two stages. During the first stage, publications related to the application of IoT in the sustainable development of smart cities are selected through screening of their titles, abstracts, and keywords. In the second stage, the full text of each selected publication is read to ensure it falls within the scope of our focus. Furthermore, we only consider the studies connecting to the real world, which can accelerate the application of the research findings. That is, the scholars may apply their research outcome to smart cities or use sensor-generated data in smart cities for data analysis. Please kindly find them on the line 191-199.
- In the descriptive analysis section, please provide more analysis to cover a wide range of relevant publications.
→ Thank you for your comments. We add more analysis in the descriptive analysis section. Please kindly find them on the line 232-240.
3: The authors should discuss how IoT technologies contribute to improving various aspects of community life, such as transportation, energy management, public safety, healthcare, and social services. Moreover, they should give more explanations of the positive impacts on sustainability, resource utilization, and citizen engagement.
→ Thank you for your comments. We discuss about smart city applications in terms of smart community, smart transportation, disaster management and emerging applications. We discussed about the application of IoT in smart community in the line 248-488. IoT can help to monitor air quality and improve waste management. We talked about the application of IoT in smart transportation in the line 489-590. IoT can improve traffic efficiency, optimize the utilization of parking lots, and improve public transport efficiency. We talked about the application of IoT in disaster management in the line 591-634. IoT can help structural health monitoring, crowd monitoring and post-disaster search and rescue. And there are some emerging applications. Please kindly find them on the line 635-654. In addition, we also highlight citizen engagement in environment monitoring on the line 291-307, and in smart transportation on the line 532-535.
4: The authors should provide more discussions on the potential risks and vulnerabilities and propose strategies to ensure data privacy, security, and ethical use of IoT technologies.
→ Thank you for your comments. We make a new section to discuss the privacy and security of the IoT system. Please kindly find them on the line 654-659.
5: More discussion is still needed to describe how IoT technologies contribute to improving transportation efficiency, reducing congestion, enhancing safety, and promoting sustainability.
→ Thank you for your comments. We develop more discussion in the section Smart Transportation. Please kindly find them on the line 488-589.
6: The authors should provide a detailed explanation of how ITS integrates various IoT technologies, such as sensors, communication networks, and data analytics, to enable real-time monitoring, control, and management of transportation systems.
→ Thank you for your comments. We develop a framework to present how ITS integrates various IoT technologies, such as sensors, communication networks, and data analytics. Please kindly find them on the line 661-669.
7: The authors also need to discuss the potential areas of innovation and research that can further enhance the role of IoT in transportation.
→ Thank you for your comment. We make a new section to discuss future research directions. Please kindly find them on the line 660-691. Especially, we present the potential areas of innovation and research that can further enhance the role of IoT in transportation on the line 678-691.
8: What about the integration of emerging technologies, such as connected and autonomous vehicles, shared mobility, and mobility-as-a-service (MaaS), and their potential impact on smart transportation?
→ Thank you for your comment. We present the integration and their potential impact on smart transportation on the line 678-691.
Reviewer 5 Report
Comments and Suggestions for Authors
The paper presents a systematic literature review for analyzing sensors on the Internet of Things Systems for the Sustainable development of Smart Cities.
The authors identify relevant research questions to which this study is dedicated. The methodology is well designed and clearly explained. The authors successfully identify 59 relevant research studies according to the methodology and conduct their analysis based on them, regarding the IoT sensors used in several relevant domains, as well as communication technologies, protocols and data integration with the IoT layer.
The paper is well written and easy to follow, as well as the methodology. However, the study does not provide sufficient details regarding the analysis performed on the identified articles. The analysis is mostly textual and enumerates all the aspects identified by the authors as important. However, a detailed analysis and synthesis of their findings is kept to a minimum.
For example, section 5.1.1 Sensors presents the findings textually. It would be much easier to follow the results if these were synthetized in a table identifying the attributes of each sensor per studied (relevant) domain.
Moreover, no details are provided regarding data captured/transmitted/stored/analyzed.
In conclusion, the study is relevant but lacks an appropriate analysis, presentation and discussion of the results. The conclusion is also not sufficiently elaborated, for such a study.
Comments on the Quality of English Language
Minor revision of the English Language used is required.
Minor remarks:
- pg.5 lines 201-204 -> paragraph seems unfinished?
- pg.5 line 201 Among --> among
- pg.8 line 329 including --> include
- pg.8 lines 339, 340 severs --> serves
- pg.8 line 345 liked --> like
- pg.9 line 394 includes --> include
etc.
Author Response
Thank you for your comments. We try to record the sensor used in each study on Appendix B. Please kindly find them on the line 727. And we completed the minor revision.
Round 2
Reviewer 1 Report
Comments and Suggestions for Authors
In this review paper, sensors in internet of things systems for the sustainable development of smart cities, the literature is systematically reviewed and the results are shared. Both versions of the paper, the previous and revised versions of the paper, have been carefully examined, and it is seen that the authors make the corrections requested by the referees and show the necessary sensitivity in the reorganizing of the paper in line with the comments. The authors have progressed in improving the paper compared to previous versions of the paper (sensors-2858047-peer-review-v1 & sensors-2858047-peer-review-v2). In the revised version of the paper, almost all the comments have been considered and addressed by the authors.
The response to reviewers is well-prepared. The changes in line with the opinions/ suggestions/ evaluations of the referees can be tracked. The existing organization problems in the previous version of the paper have been fixed. In the revised version, the clarity and follow-up of the paper have been increased.
As a result, my concerns on the previous version of the paper have disappeared with the explanations made by the authors, as well as the revision they have made.
This revision is sufficient, and it is possible to evaluate the paper for publication after necessary checks for minor spelling errors and grammar check.
Author Response
Thank you. Following your direction, we made a further improvement in the section 5,6,7. And we also improved Appendix B.
Reviewer 3 Report
Comments and Suggestions for Authors
Thanks for providing the revised manuscript. My concerns have been properly addressed.
Author Response

(The authors gave the same response as above.)

Reviewer 4 Report
Comments and Suggestions for Authors
The authors have improved the manuscript and it's ready for publication in its current form
Author Response

(The authors gave the same response as above.)

Reviewer 5 Report
Comments and Suggestions for Authors
The authors have updated their manuscript considerably. However, regarding the analysis and the conclusions, there is still need for improvement (according to previous comments). Data analysis, correlations, identified trends and future perspectives, and also aspects that must be improved in the literature are not enough clear from the current status of the paper.
Figure 3 is difficult to interpret, I suggest to redo the figure. Also, the mobile app interface in Fig 4 is not relevant to be displayed in the paper.
Comments on the Quality of English Language
English language has a good level of quality.
Author Response
Thank you for your comments. Following your direction, we made a further improvement in the section 5,6,7. We provided more future research dirctions. We revised figure 3 and removed figure 4. And we also improved Appendix B. Please kindly find in the atatchment.